# Cytoprotection by a naturally occurring variant of ATP5G1 in Arctic ground squirrel neural progenitor cells

**Neel S Singhal[1†], Meirong Bai[2,3†], Evan M Lee[2,3], Shuo Luo[2,3], Kayleigh R Cook[2,3], Dengke K Ma[2,3,4]\***

[1]Department of Neurology, University of California-San Francisco, San Francisco, United States; [2]Cardiovascular Research Institute, University of California-San Francisco, San Francisco, United States; [3]Department of Physiology, University of California-San Francisco, San Francisco, United States; [4]Innovative Genomics Institute, Berkeley, United States

**Abstract** Many organisms in nature have evolved mechanisms to tolerate severe hypoxia or ischemia, including the hibernation-capable Arctic ground squirrel (AGS). Although hypoxic or ischemia tolerance in AGS involves physiological adaptations, little is known about the critical cellular mechanisms underlying intrinsic AGS cell resilience to metabolic stress. Through cell survival-based cDNA expression screens in neural progenitor cells, we identify a genetic variant of AGS *Atp5g1* that confers cell resilience to metabolic stress. *Atp5g1* encodes a subunit of the mitochondrial ATP synthase. Ectopic expression in mouse cells and CRISPR/Cas9 base editing of endogenous AGS loci revealed causal roles of one AGS-specific amino acid substitution in mediating cytoprotection by AGS ATP5G1. AGS ATP5G1 promotes metabolic stress resilience by modulating mitochondrial morphological change and metabolic functions. Our results identify a naturally occurring variant of ATP5G1 from a mammalian hibernator that critically contributes to intrinsic cytoprotection against metabolic stress.

**\*For correspondence:**
Dengke.Ma@ucsf.edu

[†]These authors contributed equally to this work

**Competing interests:** The authors declare that no competing interests exist.

## Introduction

Arctic ground squirrels (AGS, *Urocitellus parryii*) survive harsh winter environmental conditions through hibernation. By virtue of their profound ability to suppress metabolism and core temperature, with body temperatures dropping below 0℃, AGS are known as 'extreme' hibernators (*Barnes, 1989*). Hibernation in AGS can last 7 months and is characterized by drastic (>90%) reductions in basal metabolic rate, heart rate, and cerebral blood flow (*Buck and Barnes, 2000*). Curiously, hibernation is interrupted periodically by interbout arousal (IBA) episodes in which temperature and cerebral blood flow normalize rapidly (*Drew et al., 2004*; *Karpovich et al., 2009*). Nonetheless, AGS suffer no ischemic injury during hibernation or reperfusion injury during an IBA. Hibernating ground squirrels are resistant to ischemic and reperfusion injuries in numerous models, including brain and heart tissues after cardiac arrest in vivo and hippocampal slice models derived from animals during an IBA (*Dave et al., 2009*; *Quinones et al., 2016*; *Bhowmick et al., 2017*; *Bogren et al., 2014*). This resilience to reperfusion injury does not depend on temperature of the animal or season (*Bhowmick et al., 2017*). In addition, AGS neural progenitor cells (NPCs) demonstrate resistance to oxygen and glucose deprivation ex vivo (*Drew et al., 2016*). Together, these studies suggest that in addition to physiological adaptations, AGS possess cell autonomous genetic mechanisms that contribute to intrinsic tolerance to metabolic stress or injury.

Proteomic and transcriptomic investigations have comprehensively catalogued the impact of season, torpor, and hibernation on cellular and metabolic pathways in several different tissues of

**eLife digest** When animals hibernate, they lower their body temperature and metabolism to conserve the energy they need to withstand cold harsh winters. One such animal is the Arctic ground squirrel, an extreme hibernator that can drop its body temperatures to below 0℃. This hibernation ability means the cells of Arctic ground squirrels can survive severe shortages of blood and oxygen. But, it is unclear how their cells are able to endure this metabolic stress.

To answer this question, Singhal, Bai et al. studied the cells of Arctic ground squirrels for unique features that might make them more durable to stress. Examining the genetic code of these resilient cells revealed that Arctic ground squirrels may have a variant form of a protein called ATP5G1. This protein is found in a cellular compartment called the mitochondria, which is responsible for supplying energy to the rest of the cell and therefore plays an important role in metabolic processes.

Singhal, Bai et al. found that when this variant form of ATP5G1 was introduced into the cells of mice, their mitochondria was better at coping with stress conditions, such as low oxygen, low temperature and poisoning. Using a gene editing tool to selectively substitute some of the building blocks, also known as amino acids, that make up the ATP5G1 protein revealed that improvements to the mitochondria were caused by switching specific amino acids. However, swapping these amino acids, which presumably affects the role of ATP5G1, did not completely remove the cells' resilience to stress. This suggests that variants of other genes and proteins may also be involved in providing protection.

These findings provide the first evidence of a protein variant that is responsible for protecting cells during the metabolic stress conditions caused by hibernation. The approach taken by Singhal, Bai et al. could be used to identify and study other proteins that increase resilience to metabolic stress. These findings could help develop new treatments for diseases caused by a limited blood supply to human organs, such as a stroke or heart attack.

hibernating ground squirrels, including the brain (*Quinones et al., 2016*; *Ballinger et al., 2016*; *Chang et al., 2018*; *Gehrke et al., 2019*; *Hampton et al., 2013*; *Luan et al., 2018*; *Andrews, 2019*; *Hindle et al., 2014*). Although the mechanisms underlying hibernating ground squirrel ischemia and hypothermia tolerance in the brain are not fully elucidated, studies suggest that post-translational modifications, regulation of cytoskeletal proteins, and upregulation of antioxidants play a prominent role (*Bhowmick and Drew, 2017*; *Lee et al., 2007*; *Tessier et al., 2019*). Gene expression profiling and bioinformatic analyses also indicate the cytoprotective contributions of mitochondrial and lysosomal pathways in adapting to hypothermia and hypoxia in ground squirrel and marmot species (*Bai et al., 2019*; *Ou et al., 2018*). In neurons differentiated from 13-lined ground squirrel (13LGS) induced pluripotent stem cells (iPSCs), Ou and colleagues found that hibernating ground squirrel microtubules retained stability upon exposure to hypothermia. The authors identified mitochondrial suppression of cold-induced reactive oxygen species (ROS) and preservation of lysosomal structure are key features of ground squirrel cytoprotection, and that pharmacological inhibition of ROS production or lysosomal proteases recapitulates the hypothermia-tolerant phenotype in human cells (*Ou et al., 2018*). Taken together, these studies provide important insights into pathways mediating AGS tolerance to metabolic stress. However, these studies have not focused on specific genes and proteins with cytoprotective effects uniquely evolved in hibernating ground squirrels. As such, we know very little about mechanistic details underlying genetic contribution to intrinsic stress resilience in ground squirrels.

Using a cDNA library expression-based genetic screen combined with phenotypic analyses of cell survival and mitochondrial responses to stress as compared in mouse versus AGS NPCs, we identified AGS transcripts imparting ex vivo cytoprotection against various metabolic stressors. We further use CRISPR/Cas9 DNA base editing (*Koblan et al., 2018*) to determine functional importance of amino acid substitutions uniquely evolved in AGS, and identified AGS ATP5G1[L32] as a causal contributor to stress resilience in AGS, suggesting potential for targeting this component of ATP synthase for neuroprotective treatments.

## Results

### AGS neural cells exhibit marked resistance to metabolic stressors associated with improvements in mitochondrial function and morphology

When growing under identical cell culture conditions, AGS and mouse NPCs exhibit similar morphology, growth rates and expression of Nestin and Ki67, markers for proliferating NPCs (*Figure 1A–B* and *Figure 1—figure supplement 1A-E*). Although superficially indistinguishable, mouse and AGS NPCs demonstrate markedly different responses to metabolic stressors. When exposed to hypoxia

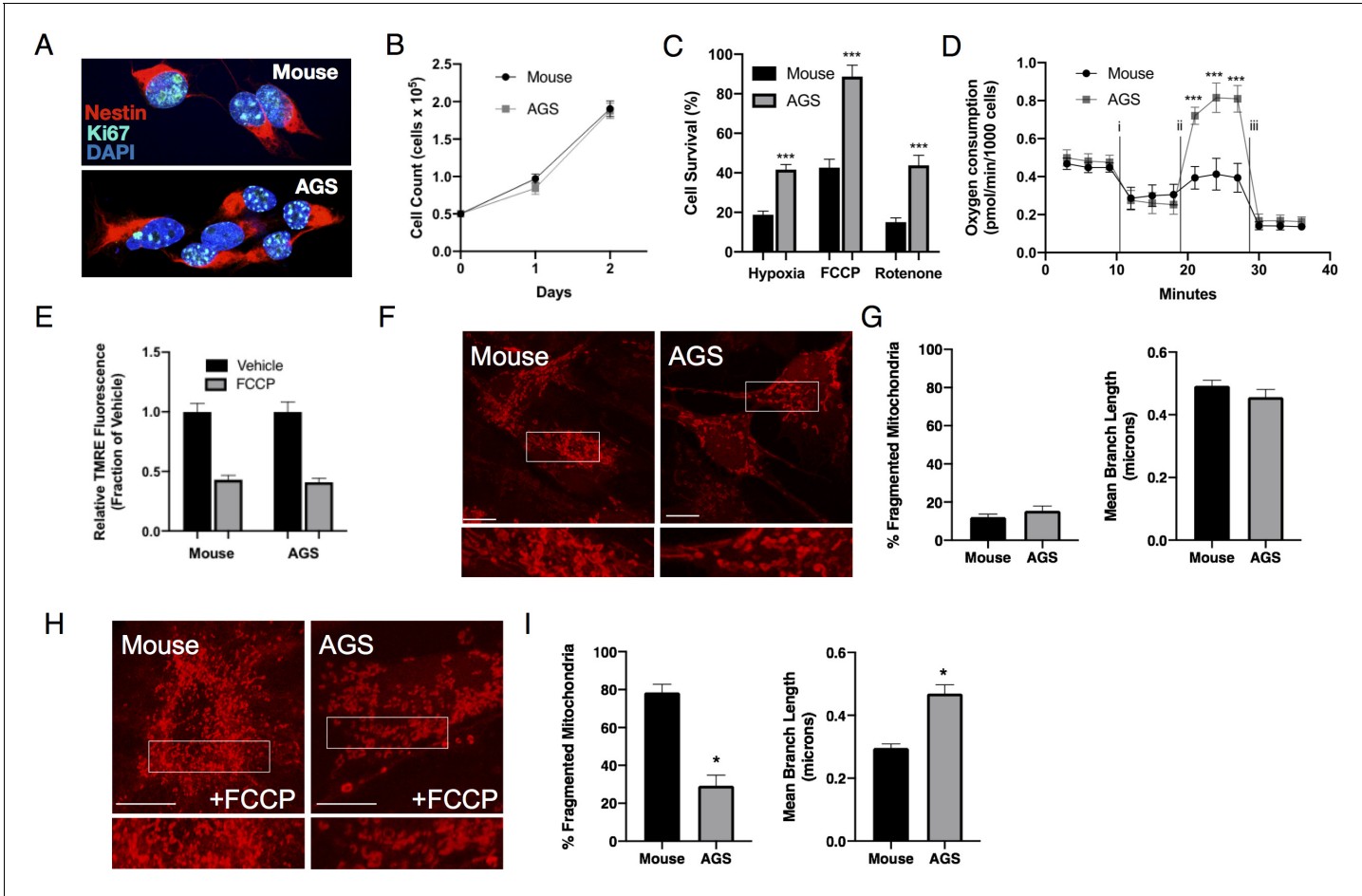

**Figure 1.** Phenotypic characteristics of Mouse and AGS NPCs. (A) Confocal image of mouse (top) and AGS (bottom) NPCs demonstrating similar morphology and expression of Nestin (red) and Ki-67 (teal) in nearly all cultured cells of both species. (B) Mouse and AGS NPCs have similar proliferation rates expressed as mean ± SEM of 3 independent experiments where 50,000 NPCs were seeded in a 24-well cell culture plate in triplicate and counted by automated cytometer on two subsequent days (C) AGS NPCs exhibit increased cell survival when exposed to hypoxia (1%, 24 hr), hypothermia (31°C, 24 hr), or rotenone (10 μM, 16 hr). Bar graphs represent the mean ± SEM of 3 independent experiments with three replicates/ condition. (D) Seahorse XF analyzer assay of cultured mouse and AGS NPCs sequentially exposed to (i) oligomycin (1 μM), (ii) FCCP (2 μM), and (iii) rotenone/antimycin (0.5 μM) showing enhanced FCCP-stimulated oxygen consumption (spare respiratory capacity). Data represents the mean ± SEM of three independent experiments with 4–6 replicates/species. (E) Relative fluorescence ± SEM of three independent experiments in triplicate each of cultured mouse and AGS NPCs loaded with TMRE (50 nM) exposed to vehicle or FCCP (1 μM) (F) Representative confocal images of mouse (left) and AGS (right) NPCs expressing the mitochondrial marker mCherry-mito7 to demonstrate mitochondrial morphology at baseline (H) and one hour following treatment with 1 μM FCCP. Scale bar represents 10 μm. (G, I) Percent of mitochondria with fragmented morphology (left panel) and the mean branch length (right panel) of mitochondrial networks of NPCs expressing mCherry-mito7. Data obtained from 30 cells/species/condition. *p<0.05; ***p<0.001.

The online version of this article includes the following figure supplement(s) for figure 1:

**Figure supplement 1.** Mouse and AGS NPCs express Nestin and Ki-67 and additional metabolic phenotypic data.

(1% O$_2$), hypothermia (31°C), or rotenone (30 µM), AGS NPCs exhibit profound resistance to cell death compared with mouse NPCs (*Figure 1C*), recapitulating resilient AGS phenotypes found in previous studies (*Dave et al., 2009*; *Bhowmick et al., 2017*; *Bogren et al., 2014*; *Drew et al., 2016*). Moreover, measurement of in vitro oxygen consumption of AGS NPCs after sequential exposure to mitochondrial toxins demonstrates strikingly higher 'spare respiratory capacity' in response to FCCP (*Figure 1D* and *Figure 1—figure supplement 1F, G*), indicating a greater metabolic reserve for stressors (*Nicholls and Budd, 2000*). Mitochondrial citrate synthase and oxidative phosphorylation (OXPHOS) enzymatic activities were similar between the two species, with the exception of complex IV (*Figure 1—figure supplement 1H*). Interestingly, functional improvements in mitochondrial function were also mirrored by changes in mitochondrial dynamic organization following exposure to FCCP at doses that lead to mitochondrial depolarization (*Figure 1E*). At baseline, mouse and AGS cells had similar mitochondrial organization as evidenced by similar mean branch length and number of cells with fragmented mitochondria (*Figure 1F*). Following FCCP treatment, mouse cells demonstrated marked increases in mitochondrial fission with concurrent decreases in mean branch length. By contrast, AGS cells appeared largely resistant to mitochondrial fission induced by FCCP (*Figure 1G*). Together, these results demonstrate intrinsic differential cell survival and mitochondrial responses to metabolic stresses between mouse and AGS NPCs.

## A cDNA library expression screen identifies AGS ATP5G1 as a cytoprotective factor

To identify cytoprotective genes expressed in AGS, we constructed a normalized cDNA expression library from AGS NPCs and introduced the library to mouse NPCs by nucleofection (*Bertram et al., 2012*; *Figure 2—figure supplement 1A-B*). Screening of inserts revealed the average library insert size was 2.4 kB. To minimize false negatives due to incorrect splice isoforms, we performed screens in triplicate and maintained representation at 1000 cells/open reading frame. Two days after AGS cDNA library nucleofection, we exposed cells to hypothermia (31°C) for 3 days, hypoxia (1%) for 2 days, or complex I inhibition (rotenone) for 3 days, respectively (*Figure 2A*). We then isolated plasmids from surviving cells, amplified cDNA insert sequences by PCR and used next-generation sequencing to identify a total of 378 putative cytoprotective genes, three of which (*Ags Atp5g1*, *Ags Manf*, and *Ags Calm1*) provided cytoprotection in all three examined metabolic stress conditions (see *Figure 2B* and *Supplementary file 1*).

Since a portion of mouse NPCs survived metabolic stresses even without AGS cDNA library expression or as a result of protective secreted factors, we anticipated false positive hits without cell autonomous cytoprotective effects. Thus, in this study, we focused on characterizing the nuclear-encoded mitochondrial protein AGS ATP5G1 that conferred cytoprotective effects independently confirmed under all three metabolic stress conditions (*Figure 2B,E–G*). ATP5G1 is one of three ATP5G isoforms making up the C-subunit of mitochondrial ATP synthases, and is regulated distinctly from ATP5G2 or ATP5G3 (*Gay and Walker, 1985*; *De Grassi et al., 2006*; *Wigington et al., 2016*). As most identified genes do not appear to be differentially expressed between mouse and AGS NPCs (*Ou et al., 2018*), we hypothesized that resistance to metabolic stress may be related to uniquely evolved AGS proteins. Based on multiple sequence alignment of the ATP5G1 protein family in mammals, we observed three AGS-unique amino acid substitutions and two small insertions/deletions at the N-terminal region of AGS ATP5G1, whereas the C-terminal membrane-spanning segment is largely invariant (*Figure 2C* and *Figure 2—figure supplement 1C*).

We expanded the analysis of AGS-unique amino acid substitutions to other cytoprotective protein variants identified from the screen of the AGS cDNA library. In particular, we analyzed uniquely evolved AGS proteins by comparing sequence alignments of the screened cytoprotective candidates for two species of ground squirrels (AGS and the 13LGS, *Ictidomys tridecemlineatus*) against nine other reference species across mammalian subclasses. We calculated the Jensen- Shannon Divergence (JSD) score, which captures sequence conservation and difference from the background amino acid distribution, and average ground squirrel-versus-other mammalian block substitution matrix (BLOSUM)−62 scores for each unique residue (*Capra and Singh, 2007*). High JSD and low BLOSUM62 scores indicate chemically significant amino acid substitutions, and as such potentially important functional AGS adaptations. We found that the leucine-32 residue of AGS ATP5G1 in place of the otherwise highly conserved proline is unique to hibernating ground squirrels, and on conservation analysis scored among the highest of all AGS-unique amino acid substitutions in

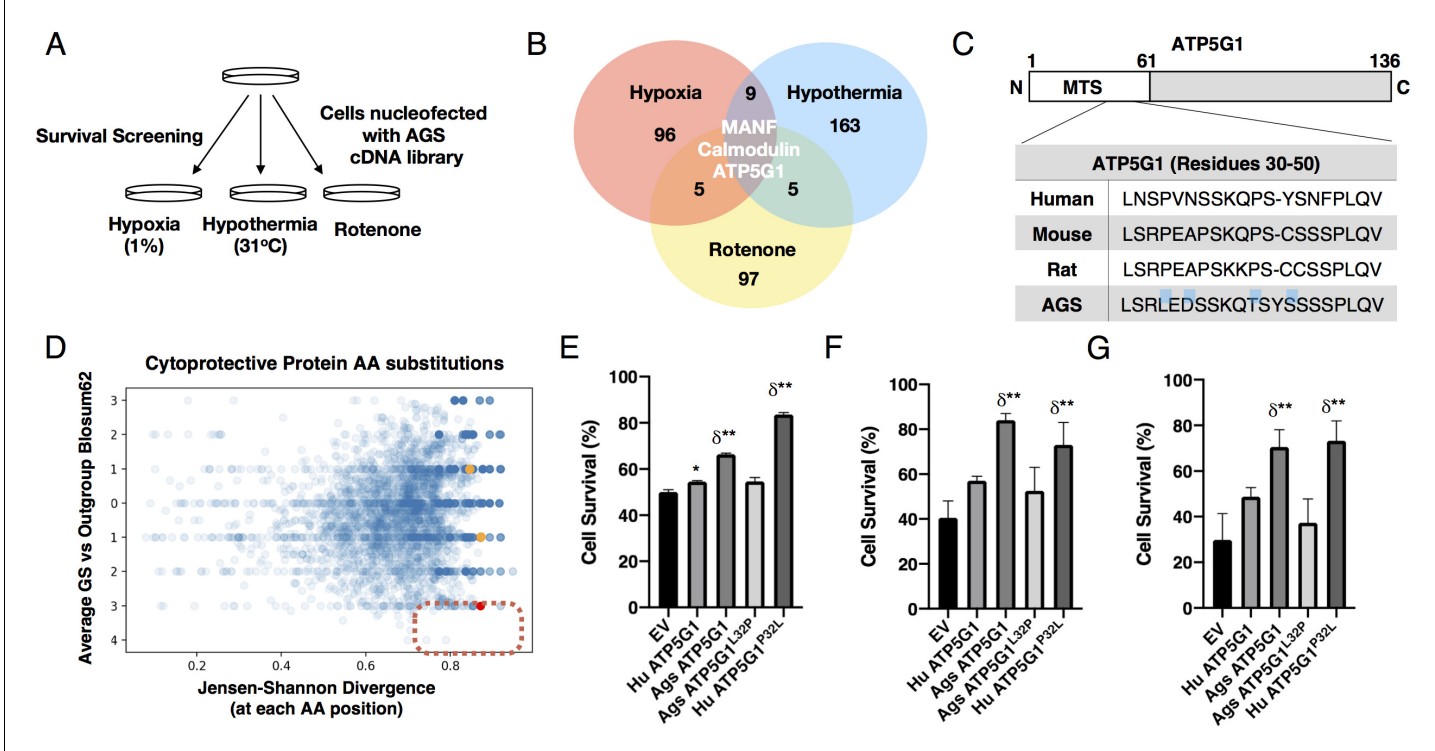

**Figure 2.** AGS cDNA library survival screen identifies AGS ATP5G1 as a cytoprotective factor. (**A**) AGS NPC cDNA was introduced into mouse NPCs by nucleofection. Cells were screened for survival after exposure to hypoxia (1%, 48 hr), hypothermia (31°C, 72 hr), or rotenone (20 µM, 48 hr) to identify AGS cytoprotective factors. (**B**) Venn-diagram demonstrating the number of cytoprotective proteins identified by next-generation sequencing of plasmids isolated from cells surviving each condition of the cDNA library screen. (**C**) Truncated sequence alignments demonstrating key GS AA substitutions (blue highlight) for ATP5G1, one of the three proteins imparting survival in all three screens. (**D**) Ground squirrel-unique amino acid substitutions are plotted as a function of BLOSUM62 score and Jensen-Shannon Divergence (JSD) score. Ground squirrel-unique AA substitutions with the highest probability of functional consequence are in the denoted red quadrant (top 1% scoring of high JSD values and low BLOSUM62 scores). The red dot represents the ATP5G1$^{L32P}$ substitution; orange dots represent two other ATP5G1 substitutions. (**E–G**) Mouse NPCs expressing human ATP5G1, AGS ATP5G1, AGS ATP5G1$^{L32P}$, human ATP5G1 $^{P32L}$, or empty vector (EV) and exposed to 24 hr of 1% $O_2$ (**E**), 31°C (**F**), or 20 µM rotenone (**G**). Cell death was determined by flow cytometry for propidium iodide and experiments are mean ± SEM of three independent experiments with three replicates/genotype/condition, *$p < 0.05$ or **$p < 0.01$ vs EV; δ $< 0.05$ vs human ATP5G1.

The online version of this article includes the following figure supplement(s) for figure 2:

**Figure supplement 1.** cDNA library construction and ATP5G expression in mouse and AGS NPCs.

identified cytoprotective protein candidates from our screen (*Andersson et al., 1997*; *Figure 2D* and *Supplementary file 2*).

The N-terminal region of ATP5G proteins can undergo cleavage, but also modulate mitochondrial function directly, by unknown mechanisms (*Vives-Bauza et al., 2010*). Although the three C-subunit proteins are identical in sequence, they cannot substitute for one another and are all required to constitute a fully functional C-subunit (*Vives-Bauza et al., 2010*; *Sangawa et al., 1997*). To determine the relative levels of ATP5G1, −2, and −3 in mouse and AGS NPCs, we performed qRT-PCR analysis with species and transcript-specific primers. We found that in both mouse and AGS NPCs, expression of *Atp5g3* or *Atp5g2* is greater than that of *Atp5g1*, consistent with prior reports in human and mouse tissues (*Gay and Walker, 1985*; *Vives-Bauza et al., 2010*). We found that the relative abundance of the *Atp5g1* isoform is elevated nearly twofold in AGS NPCs (*Figure 2—figure supplement 1D*). However, the relative abundance of the mature ATP5G (subunit C) protein or oligomycin sensitivity of complex V activity, is not different in mouse and AGS cells (*Figure 2—figure supplement 1E-F*).

Overexpression of the AGS variant of ATP5G1 in mouse NPCs confers cytoprotection in cells exposed to hypoxia, hypothermia, or rotenone (*Figure 2E–G*). We found that this protective response is not present in NPCs overexpressing ATP5G1$^{L32P}$. Conversely, overexpression of the

human ATP5G1$^{P32L}$, which mimics the wild-type AGS ATP5G1 variant, leads to enhanced cytoprotection in these conditions of metabolic stress compared to that of human ATP5G1. The ATP5G1 substitutions did not alter the mitochondrial localization of ATP5G1 when expressed in either mouse or AGS NPCs (*Figure 3—figure supplements 1,2*). In addition, overexpression of the AGS variant of ATP5G1 recapitulated key features of the AGS resilient mitochondrial phenotype, including increasing spare respiratory capacity and reducing mitochondrial fission with reduced fragmentation and

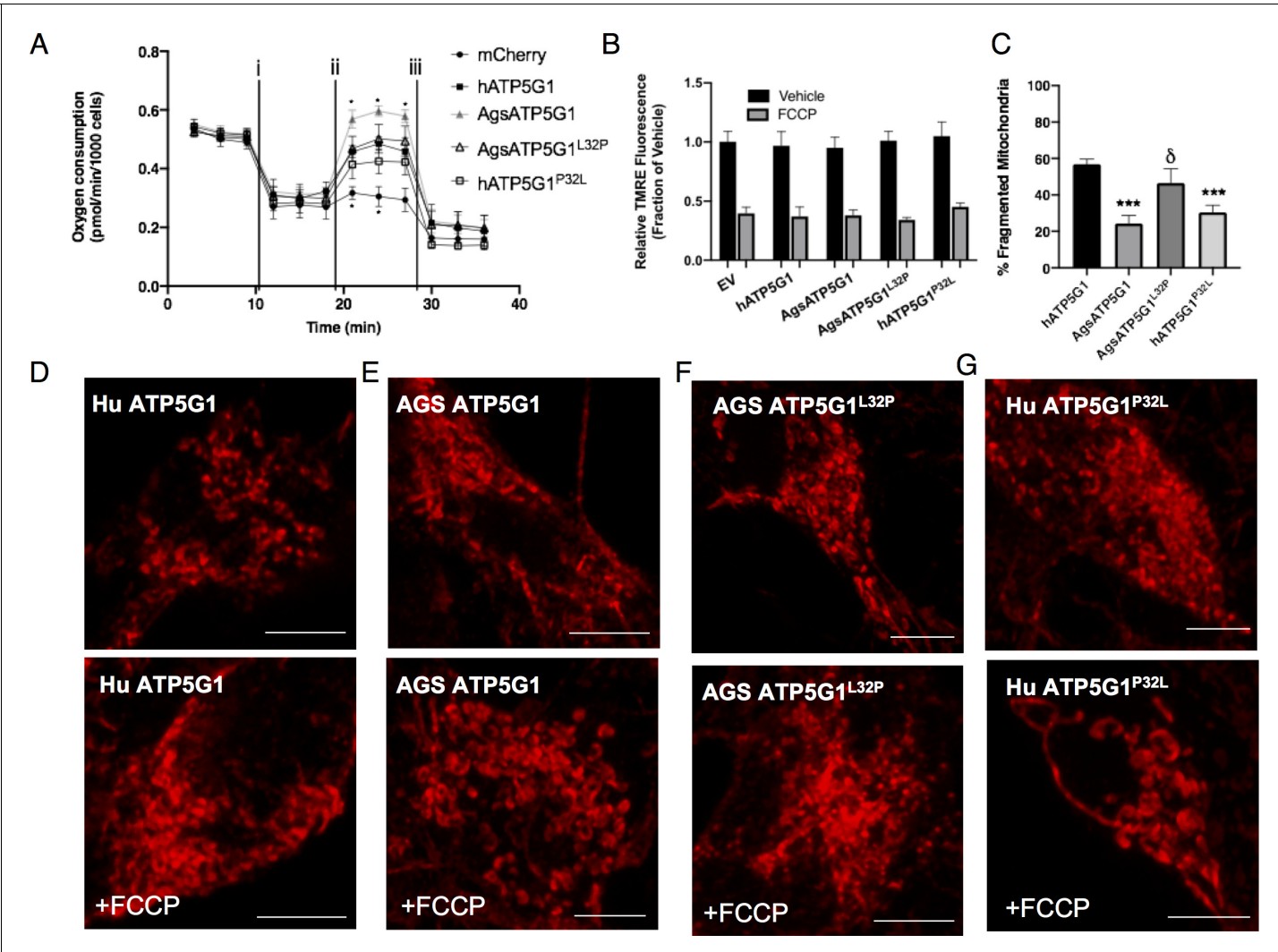

**Figure 3.** Overexpression of AGS ATP5G1 in mouse NPCs recapitulates AGS metabolic phenotypes, which is dependent on the uniquely evolved leucine-32. (**A**) Seahorse XF analyzer assay of cultured mouse NPCs expressing human ATP5G1, AGS ATP5G1, AGS ATP5G1$^{L32P}$, human ATP5G1$^{P32L}$, or empty vector and sequentially exposed to (i) oligomycin (1 μM), (ii) FCCP (2 μM), and (iii) rotenone/antimycin (0.5 μM) showing increased FCCP-stimulated oxygen consumption (spare respiratory capacity) with AGS ATP5G1. Substitution of the AGS leucine-32 results in reduced spare respiratory capacity. Data represents the mean + SEM of three independent experiments with 4–6 replicates/species. (**B**) Relative fluorescence ± SEM of three independent experiments in triplicate each of cultured mouse NPCs stably expressing the indicated ATP5G1 construct, loaded with TMRE (50 nM), and exposed to vehicle or FCCP (1 μM) (**C**) Percent ± SEM fragmented mitochondria and representative confocal images (**D-G**) of mitochondrial networks in mouse NPCs expressing human, AGS, and mutant forms of mCherry-ATP5G1 one hour following treatment with vehicle (top panel) or 1 μM FCCP (bottom panel). Data obtained from 30 cells/condition. *p<0.05; ***p<0.001 vs. human ATP5G1; δ <0.05 vs AGS ATP5G1. Scale bars represent 5 μm. The online version of this article includes the following figure supplement(s) for figure 3:

**Figure supplement 1.** Additional data for overexpression of AGS ATP5G1 in mouse NPCs recapitulating AGS metabolic phenotypes, which is dependent on the uniquely evolved leucine-32.

**Figure supplement 2.** Human and AGS ATP5G1 constructs are appropriately targeted to the mitochondria of mouse NPCs.

**Figure supplement 3.** Human and AGS ATP5G1 constructs are appropriately targeted to the mitochondria of AGS NPCs.

increased branch length of mitochondria in response to FCCP (*Figure 3*, *Figure 3—figure supplement 3A*). Interestingly, NPCs expressing the AGS ATP5G1$^{L32P}$ variant demonstrated reduced spare respiratory capacity and increased mitochondrial fragmentation compared to the AGS ATP5G1 over-expressing NPCs. Overexpression of human ATP5G1$^{P32L}$ improved survival to metabolic stressors and reduced mitochondrial fragmentation, but compared to AGS ATP5G1$^{L32P}$ spare respiratory capacity was not significantly improved. This may indicate that improving spare respiratory capacity itself is not the sole mechanism conferring resilience to metabolic stressors. Of note, expression of AGS ATP5G1 with two other identified AGS-unique amino acid substitutions (N34D, T39P) did not affect survival of mouse NPCs exposed to hypoxia, hypothermia, or rotenone (*Figure 3—figure supplement 3B–D*). Together, these results reveal cytoprotective effects of AGS *Atp5g1* when ectopically expressed in metabolic stress-susceptible mouse NPCs, and identify functional importance of the leucine-32 residue of AGS ATP5G1 uniquely evolved in AGS.

## Knock-in of AGS ATP5G1$^{L32P}$ alters the resilient phenotype of AGS cells

Species-specific substitutions of amino acid residues at sites deeply conserved in mammals indicate either relaxed selective constrains at the sites during evolution or potentially adaptive significance functionally specific for that species. As ectopic expression may not fully reflect endogenous functions, precise manipulation of endogenous genetic loci is required to determine definitive causal contribution of ATP5G1$^{L32}$ to the metabolic resilience of AGS. Using the recently reported adenine DNA base editor (ABEmax; 22), we successfully generated AGS cell lines homozygous for ATP5G1$^{L32P}$ by introducing a cytosine-to-thymine substitution in the (-) strand of *Ags Atp5g1* (*Figure 4A,B*). We isolated three clonal AGS NPC lines harboring the desired knock-in mutation (ABE KI) and two clonal lines that underwent editing and remained homozygous for the wild-type allele (ABE WT). Compared to ABEmax-treated AGS cells without successful knock-in (*Figure 4—figure supplement 1A*), ABE KI cell lines did not demonstrate differences in *Atp5g1* mRNA expression, protein abundance, or complex V activity (*Figure 4—figure supplement 1B-C*, *Figure 4I*). However, knock-in of the L32P residue resulted in markedly reduced survival of AGS NPCs following exposure to hypoxia, hypothermia, or rotenone (*Figure 4C*). In addition, we found the ABE KI AGS NPCs exhibited marked reduction in 'spare respiratory capacity' and altered mitochondrial dynamics in response to FCCP treatment (*Figure 4D–H* and *Figure 4—figure supplement 1C*). Although overall ATP5G protein abundance is unchanged (*Figure 4—figure supplement 1D-E*), we used clear-native gel electrophoresis (*Kovalčíková et al., 2019*; *Wittig and Schägger, 2009*) and identified a reduced presence of ATP synthase dimers relative to the total amount of ATP synthase in ABE KI cells (*Figure 4J–K*). Further biochemical experiments are necessary to delineate the specific mechanisms of how the AGS leucine-32 substitution affects the assembly or stability of ATP synthase complex proteins. Nonetheless, genetic evidence in our study based on ectopic expression and specific CRISPR base editing of endogenous loci demonstrates causal roles of the AGS leucine-32 substitution in cytoprotection. Collectively, these results identify a naturally occurring cytoprotective AGS variant that contributes to cytoprotection against various metabolic stresses likely by modulating mitochondrial function.

## Discussion

Previous studies have indicated that hibernating organisms evolved numerous physiological and cellular mechanisms enabling survival during the stressed metabolic conditions accompanying hibernation (*Bai et al., 2019*; *Ou et al., 2018*; *Ballinger et al., 2017*). However, we still know little about the mechanistic details of how AGS protein-coding genetic variants contribute to intrinsic cytoprotective functions. We show that ex vivo cultured AGS NPCs can recapitulate remarkable intrinsic resilience to hypoxia, hypothermia, and other metabolic stressors. Additionally, using an unbiased cDNA expression screening and bioinformatic strategy, we identified numerous AGS transcripts and uniquely evolved AGS amino acid substitutions potentially contributing to cytoprotection. We focused on discerning the protective effect of AGS ATP5G1, a nuclear-encoded mitochondrial protein, given that it was one of only three genes identified in all three metabolic stress paradigms and the prominent mitochondrial resilience phenotype of AGS NPCs. We hypothesize that analogous to amino acid substitutions in several human proteins providing adaptive benefits (*Simonson et al., 2010*; *Song et al., 2014*; *Xiang et al., 2013*; *Yates and Sternberg, 2013*), substitutions in AGS

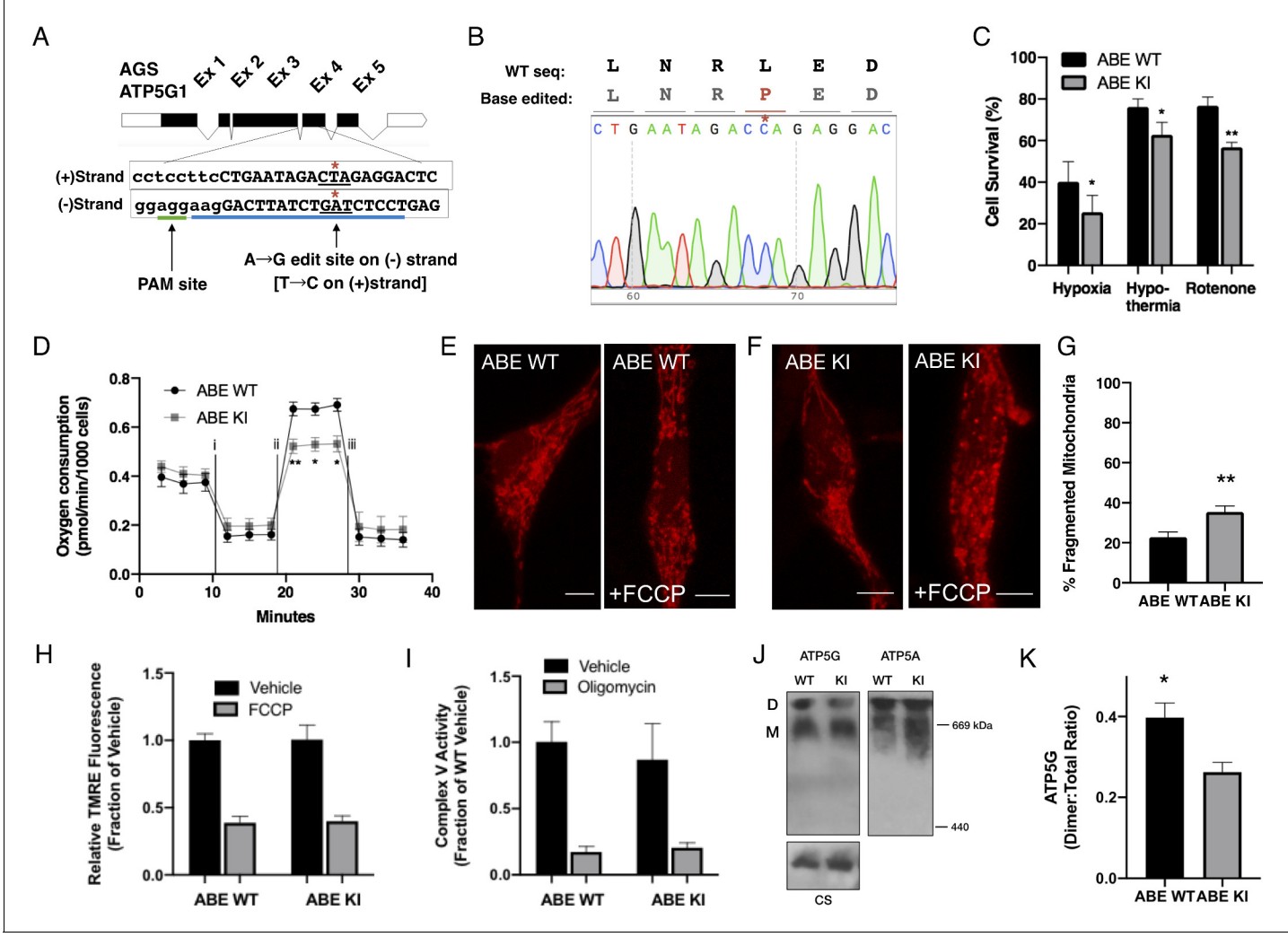

**Figure 4.** CRISPR base editing to generate ATP5G1[L32P] AGS NPCs results in a partial loss of AGS metabolic resilient phenotypes. (**A**) AGS ATP5G1 CRISPR base editing strategy. To create AGS cells with the human amino acid substitution at leucine-32, AGS cells transiently expressing ABEmax were nucleofected with an sgRNA (blue underline) directed toward a PAM site (green underline) on the (-) strand to target conversion of adenine to guanine, which on the (+) strand is a cytosine-to-thymine (*). (**B**) Sequencing data from a successfully edited clonal AGS cell line demonstrating the cytosine-to-thymine base edit resulting in the desired leucine to proline knock-in cell line. (**C**) AGS ATP5G1[L32P] (ABE KI) NPCs exhibit decreased cell survival compared to unedited AGS cells (ABE WT) when exposed to hypoxia (1%, 24 hr), hypothermia (31°C, 72 hr), or rotenone (10 µM, 16 hr). Bar graphs represent the mean ± SEM of three independent experiments with three replicates/condition. (**D**) Seahorse XF analyzer assay of cultured ABE KI and WT cells sequentially exposed to (i) oligomycin (1 µM), (ii) FCCP (2 µM), and (iii) rotenone/antimycin (0.5 µM) showing enhanced FCCP-stimulated oxygen consumption (spare respiratory capacity). Data represents the mean ± SEM of 3 independent experiments with 4–6 replicates/species. (**E and F**) Representative confocal images of ABE WT (**E**) and ABE KI (**F**) NPCs expressing the mitochondrial marker mCherry-mito7 to demonstrate mitochondrial morphology one hour following treatment with FCCP. Scale bar represents 5 µm. (**G**) Percent of mitochondria ± SEM with fragmented morphology, data obtained from 50 to 60 cells/genotype. (**H**) Relative fluorescence ± SEM of 3 independent experiments in triplicate each of cultured ABE AGS NPCs loaded with TMRE (50 nM) and exposed to vehicle or FCCP (1 µM). (**I**) Complex V enzymatic activity normalized to protein content of ABE WT and KI mitochondrial extracts normalized to protein content and treated with vehicle or oligomycin (1 µM). Data are the mean ± SEM of 3 independent experiments expressed as a fraction of ABE WT enzymatic activity. (**J**) Representative immunoblots for ATP5G (left), ATP5A (right), or citrate synthase (CS, left, input control) of clear-native gel electrophoresis of mitochondrial extracts from ABE WT and ABE demonstrate ATP synthase dimers (**D**) and monomers (**M**). (**K**) Quantification of ATP5G demonstrates a reduction in ATP synthase dimers relative to total ATP synthase protein (D:(D+M) ratio) in ABE KI. Data are mean ± SEM of 3 independent blots. *p<0.05; **p<0.01.

The online version of this article includes the following figure supplement(s) for figure 4:

**Figure supplement 1.** Additional data for ABE cell lines.

ATP5G1 may underlie AGS adaptive mechanisms contributing to its robust cytoprotective phenotype. Using the dCas9 ABE technology, we validated a unique AGS ATP5G1$^{L32}$ amino acid substitution in the N-terminal region of ATP5G1 that leads to improvements in mitochondrial physiologic parameters. Thus, our study used CRISPR base editing in non-model organism hibernator cells, for the first time to our knowledge, to identify a naturally occurring cytoprotective protein variant from AGS. CRISPR edited ATP5G1$^{L32P}$ did not fully abolish the metabolic resilience phenotype in AGS NPCs, indicating that other gene variants may also be involved. The robust ex vivo paradigm of AGS phenotypes established from our study makes it tractable to investigate additional gene and protein variants that contribute to the metabolic resilience phenotype in AGS. Further understanding the gene variants and mechanisms responsible for the AGS phenotype has important implications for novel neuroprotective treatments in ischemic diseases as well as promoting survival of neural stem cell grafts (*Bernstock et al., 2017*).

Mitochondrial metabolic dysfunction is central to ischemia and reperfusion injury. Physiologic, transcriptomic, and proteomic studies have highlighted the importance of ketone and fatty acid metabolism in hibernating states (*Brown and Staples, 2014*; *Xu et al., 2013*) as well as pointed to a role for specific post-translational protein modifications in the differential regulation of metabolic pathways in hibernation (*Ballinger et al., 2016*; *Chung et al., 2013*; *Herinckx et al., 2017*). Specific variants of neuroprotective proteins have also been identified to be upregulated in ground squirrels during hibernation including s-humanin, however, the phenotypic or mechanistic consequences of these variants are not known (*Szereszewski and Storey, 2019*). We expanded this body of knowledge, by identifying altered mitochondrial dynamics and enhanced spare respiratory capacity in cells of AGS as potentially adaptive cellular mechanisms in hibernating animals. This mitochondrial phenotype is likely responsible for the broad resilience of AGS cells against a wide range of metabolic stressors.

Spare respiratory capacity, as measured by FCCP-stimulated oxygen consumption, represents a marker for cellular metabolic reserves, correlates with metabolic resilience (*Nicholls and Budd, 2000*) and is thought to be determined by the oxidative phosphorylation machinery (*Pfleger et al., 2015*; *Yadava and Nicholls, 2007*). Notably, human and mouse NPCs and neural cells have been reported to have diminished spare respiratory capacity as they may respire maximally at baseline (*Khacho et al., 2016*; *Lorenz et al., 2017*). However, AGS demonstrate marked elevations in spare respiratory capacity compared to mouse cells, which likely explains marked AGS NPC survival even under complex I inhibition by rotenone (*Yadava and Nicholls, 2007*). While the elevated spare respiratory capacity is likely the result of AGS adaptations in numerous metabolically active proteins, the importance of the ATP5G1 variant is highlighted by our experimental evidence demonstrating improvement in spare respiratory capacity in mouse NPCs over-expressing AGS ATP5G1 variants and decreased spare respiratory capacity in AGS NPCs with ATP5G1 L32P knock-in. A critical role for ATP5G1 in cellular energetics is also supported by recent work uncovering ATP5G1 as one of the major effectors of the transcription factor, BCL6, in regulating adipose tissue energetics as well as maintaining thermogenesis in response to hypothermia (*Kutyavin and Chawla, 2019*; *Senagolage et al., 2018*).

Mitochondrial fission and fusion are regulated by cellular metabolic state and a host of regulatory proteins, many of which have been implicated in cell survival response to stresses (*Labbé et al., 2014*). While metabolic stresses often lead to mitochondrial fission followed by apoptosis, mitochondrial fusion and resistance to fission in response to stress are anti-apoptotic (*Abdelwahid et al., 2007*; *Chen et al., 2007*). Fusion is hypothesized to allow for complementation of damaged and dysfunctional mitochondria, and in states of metabolic stress, hyperfusion of mitochondria helps maintain mitochondrial membrane potential and cell viability (*Gomes et al., 2011*). The increase in fusion and improvement in cell survival in mouse NPCs over-expressing AGS ATP5G1 and loss of resilient metabolic phenotypes in AGS cells carrying ATP5G1$^{L32P}$ underscore the importance of this pathway in altering mitochondrial morphologic response to metabolic stresses and increasing the metabolic oxidative capacity of cells.

In mammals, many of the approximately 1000 nuclear-encoded mitochondrial proteins contain a unique mitochondrial targeting sequence (MTS) providing a high degree of specificity in regulating mitochondrial import and sorting. These mitochondrial targeting and processing functions are regulated by the highly conserved mitochondrial membrane translocating protein complexes (TOM and TIM) and MTS cleaving proteins, mitochondrial processing peptide (MPP) and mitochondrial

intermediate peptide (MIP). Processing of ATP5G1 and its incorporation of the mature peptide into oligomeric c-rings and Complex V-Fo appear to involve cleavage by MPP and stabilization by TMEM70 (*Kovalčíková et al., 2019*). We did not find evidence that the ATP5G1 MTS sequence variations from AGS and human/mouse affected the mitochondrial localization or cleavage of the immature protein. This is likely due to evolutionarily conserved mitochondrial import sequence motifs and the putative ATP5G1 MPP/MIP cleavage site (xRx↓(F/L/I)xx(S/T/G)xxxx↓; see *Figure 2—figure supplement 1C*; *Gakh et al., 2002*). Interestingly, under native gel electrophoresis conditions in ABE KI NPC mitochondria, we observed a reduction in ATP synthase dimers relative to total ATP synthase. Additional supporting evidence is required to understand the mechanistic basis of this effect. We speculate that the AGS variant alters ATP5G1 processing which subsequently affects downstream dimerization of ATP synthases. Suprastructural alterations in ATP synthase organization are known to be critical to mitochondrial morphology and formation of the mitochondrial permeability transition pore (MPTP) (*Nesci and Pagliarani, 2019*). Though the exact nature of the relationship between ATP5G1 and the MPTP is controversial, many studies demonstrate improved bioenergetic responses and cell survival with ATP synthase dimerization (*Bonora et al., 2017*; *Daum et al., 2013*; *García-Aguilar and Cuezva, 2018*). Others have postulated that the cleaved ATP5G1 N-terminal mitochondrial targeting sequence modulates mitochondrial function downstream of Complex IV distinct from the functionally active C-terminal protein (*Vives-Bauza et al., 2010*). Although increased abundance of the *Atp5g1* transcript in AGS compared with mouse NPCs could contribute to the altered mitochondrial function as in prior investigations of regulation of ATP synthase in mouse brown adipose tissue (*Andersson et al., 1997*), the ABE ATP5G1$^{L32P}$ KI cells did not demonstrate a difference in *Atp5g1* mRNA transcript abundance, further supporting the notion that AGS ATP5G1$^{L32P}$ contributes to cytoprotection likely via post-transcriptional processing of ATP5G1. Precise mechanisms of how AGS ATP5G1$^{L32P}$ affects mitochondrial function and metabolic stress resilience phenotypes await future investigations.

Further unraveling of the mechanisms underlying AGS mitochondrial and cellular resilience to metabolic stress or injuries holds the hope of finding novel cytoprotective strategies that may lead to improved treatments for human diseases. Systematic investigation of additional cytoprotective genes and amino acid substitutions identified from AGS should provide important insights into the mechanism and pathways underlying intrinsic stress resilience to metabolic stresses. The use of CRISPR gene editing technologies coupled with phenotypic analysis in AGS NPCs is a new and powerful approach to evaluate causal roles of genetic variants in conferring phenotypic traits of AGS traditionally intractable to study. Identification and analysis of such causal variants for stress resilience in AGS may help develop pharmacological, gene therapy, or CRISPR/genome editing-based therapeutic strategies to treat human ischemic disorders, including stroke and heart attack.

## Materials and methods

### Cell culture

AGS NPCs (Neuronascent, Gaithersburg, MD, USA) and mouse NPCs (gift of Song lab, Baltimore, MD) have been previously described (*Drew et al., 2016*; *Ma et al., 2009*). They were grown under standard conditions at 37°C and 5% $CO_2$ with NeuroCult basal media (STEMCELL, Vancouver, BC, CA) with EGF (50 ng/ml, PeproTech, Inc, Rocky Hill, NJ, USA), FGF (100 ng/ml, PeproTech, Inc), heparin (0.002%), and proliferation supplements (STEMCELL). Early passage cultures (P2) were expanded and frozen and thawed in batches for use in experiments. These cultures contain cells ubiquitously expressing the NPC marker, Nestin, and the proliferation marker, Ki-67 (*Figure 1—figure supplement 1*). For in vitro modeling of metabolic stress, cells were exposed to either: (i) 1% hypoxia in a specialized incubator (Nuaire, Plymouth, MN, USA) saturated with Nitrogen/5% $CO_2$; (ii) hypothermia in standard incubators maintained at lower temperatures; and (iii) complex I inhibition with the addition of rotenone to cell media. For cell proliferation determination, wells were seeded in triplicate with 50,000 cells. On subsequent consecutive days, cells were detached with Accutase (STEMCELL) and counted by automated cytometry (Nanoentek, Waltham, MA, USA).

## DNA constructs and lentiviral transfection

The pHAGE-ATP5G plasmids were generated by direct PCR and PCR fusions; and the point mutation plasmids generated using Q5 site-directed mutagenesis (New England Biolabs, Beverly, MA, USA). For lentiviral transfection, the plasmids with packaging plasmids were co-transfected into HEK293FT (with a ratio of 2:1.5:1.5) using Turbofect reagent (Thermo Fisher Scientific Inc, Waltham, MA, USA) according to the manufacturer's instructions. Lentivirus-containing medium was filtered from the post-transfection supernatant and used for transduction of HEK293T cells or mouse NPCs. All lentivirus-infected cells were cultured in the medium containing Polybrene (4 µg/ml; Sigma Aldrich, St. Louis, MO, USA) for 8 hr before changing media. Forty-eight hours after transduction, the cells were selected with 10 µg/ml Blastidicin S (Thermo Fisher Scientific Inc).

## Generation of CRISPR base-edited AGS cells

ATP5G1$^{L32P}$ NPCs were generated using the dCas9 base editor, ABEmax (gift from David Liu, Addgene #112095), as previously described (*Koblan et al., 2018*). Briefly, a synthetic sgRNA (TCCTCTAG TCTATTCAGGAA) was selected by manual inspection of the AGS *Atp5g1* sequence for a PAM (NGG) site near the desired edit on the (-) strand of the gene. AGS NPCs were nucleofected (Amaxa 4D, program DS113) in P3 solution (Lonza, Alpharetta, GA, USA) containing pCMV ABEmax (500 ng/ 200,000 cells). Following a 48 hr recovery period, the same cells were nucleofected with the synthetic sgRNA sequence above (100 pmol, Synthego, Menlo Park, CA, USA). Cells were expanded and then clonally plated. Clones were screened by PCR as the desired base edit also introduced a new BfaI restriction enzyme cutting site. Sanger sequencing was used to confirm the two WT and three KI clone sequences utilized. Potential off-target effects of CRISPR/Cas9 cleavage were analyzed by Sanger sequencing of the top 5 predicted off-target genomic locations [https://mit.crispr. edu], which demonstrated a lack of indels for all clones used in subsequent analysis.

## Cell death assay

Mouse and AGS cells were plated in 24 or 96-well plates and grown to 70% confluence. Cells were exposed to metabolic stress paradigms as above, and detached and floating cells collected by centrifugation and washed with 1 ml PBS. The collected cells were resuspended with 200 µl PBS with addition of 0.2 µl Sytox blue (1 µM; Thermo Fisher Scientific) or propidium iodide (2 µg/ml) for an additional 5 min. The fluorescence intensity was measured for individual cells using automated cytometry (Nanoentek) or flow cytometry (BD Biosciences, San Jose, CA, USA) within 20 min of staining, and the percentage of cell death quantified using the FlowJo software.

## cDNA Library generation, screening, and identification of AGS amino acid substitutions

RNA was isolated from AGS NPC cells grown under standard conditions. A normalized cDNA library was generated by a commercial research partner (Bio S and T, Montreal, QC, Canada) from RNA extracted from AGS NPCs. Library quality and normalization is shown in *Figure 2—figure supplement 1A and B*. For library screening, plates containing $1 \times 10^7$ mouse NPCs cells were grown in triplicate and nucleofected with 200,000 clones each. Plates were exposed to one of three metabolic stress conditions (hypoxia, hypothermia, or rotenone treatment) for 48–96 hr. Following this treatment, plasmid DNA was purified from surviving cells and PCR-amplified AGS cDNA inserts subjected to next-generation sequencing. Resulting fastq files were trimmed (Trim Galore!) and mapped to the Ictidomys Tridecemlineatus genome (SpeTri2.0) using HISAT2. Mapped reads were subjected a custom pipeline for analyzing amino acid substitutions (https://github.com/evanmlee/MaLab_spec_ subs; copy archived at https://github.com/elifesciences-publications/MaLab_spec_subs; *Singhal, 2020*). Briefly, protein sequences of mapped genes were queried by gene symbol and downloaded from OrthoDBv10 for 10 species (13LGS, *Mus musculus*, *Rattus norvegicus*, *Sorex araneus*, *Pongo abelii*, *Homo sapiens*, *Equus caballus*, *Bos taurus*, *Oryctolagus cuniculus*, *Sus scrofa*). OrthoDB data was filtered by matching records against accepted GeneCards aliases for each gene (*Kriventseva et al., 2019*). Multiple records per species were resolved using maximum percent identity against the accepted human, mouse, and 13LGS sequences, such that only one record per species was used for alignment. AGS protein sequences were downloaded from the Entrez Protein database. Multiple AGS isoforms were resolved by best identity match to the OrthoDB sequence

data. The final protein sequence set was aligned with KAlign 2.04 (*Lassmann and Sonnhammer, 2005*). From aligned sequences, GS-specific residue substitutions were defined as amino acid variants present in 13LGS and AGS sequences and present in no other included species. For each GS-specific residue, sequence weights, JSD, and average GS-versus-outgroup BLOSUM62 scores were calculated as described previously (*Capra and Singh, 2007*). BLOSUM62 scores were used instead of point-accepted mutation scores in order to prioritize protein sequence changes with higher probability of potential chemical and functional difference. JSD was used to capture sequence conservation and difference from the background amino acid distribution. BLOSUM62 scores were calculated for GS residues against all other mammalian species sequences and averaged to give GS vs Outgroup BLOSUM62. For the entire screened cytoprotective protein dataset, JSD and BLOSUM62 score were plotted for individual genes of interest against the remaining dataset.

## Analysis of in vitro mitochondrial respiration

Analysis of mitochondrial respiratory potential was performed using a flux analyzer (Seahorse XF$^e$96 Extracellular Flux Analyzer; Seahorse Bioscience, North Billerica, MA, USA) with a Seahorse XF Cell Mito Stress Test Kit according to the manufacturer's instructions. Basal respiration and ATP production were calculated to evaluate mitochondrial respiratory function according to the manufacturer's instructions. After the measurement, cells were harvested to count the cell number, and each plotted value was normalized relative to the number of cells used. Briefly, NPCs were seeded (25,000 cells/well) into each well of XF$^e$96 cell culture plates and were maintained in standard culture media. After 2–3 days in culture, cells were equilibrated in unbuffered XF$^e$assay medium (Seahorse Bioscience) supplemented with glucose (4.5 g/L), sodium pyruvate (25 mg/L) and transferred to a non-CO$_2$ incubator for 1 hr before measurement. Oxygen consumption rate (OCR) was measured with sequential injections of oligomycin, FCCP, and rotenone/antimycin A.

## Analysis of mitochondrial respiratory chain complex activity and mitochondrial potential

Analysis of mitochondrial respiratory chain complex I, II, and IV activity was measured in mitochondrial extracts using complex enzyme activity colorimetric or absorbance-based assays (ab109721, ab10908, ab109911; Abcam, Cambridge, MA). Complex V activity was measured with Complex V Mitocheck kit (Cayman Chemical, Ann Arbor, MI, USA) and citrate synthase activity with a Citrate Synthase Enzyme Assay (Detroit R and D, Detroit, MI). Mitochondrial extracts (50 µg) were obtained as previously described (*Clayton and Shadel, 2014*) and used to measure time-dependent absorbance alterations on a multi-well plate reader (SprectraMax, Molecular Devices, San Jose, CA, USA). Mitochondrial membrane potential was evaluated by loading $1 \times 10^5$ cells in triplicate with the lipophilic positively charged dye tetramethylrhodamine ethyl ester (TMRE, 50 nM). For depolarization control wells, 1 µM FCCP was added. Excitation and emission wavelengths (530 and 580 nm, respectively) were measured on a multi-well plate reader.

## Mitochondrial ATP5G1 targeting and dynamic morphology assessment

Mitochondrial localization of ATP5G1 constructs as well as morphology and fission/fusion is assessed in mouse and AGS NPCs nucleofected with mCherry or mEmerald-mito7 (Gift from Michael Davidson, Addgene #55102, 54160) as a mitochondrial marker and grown on glass coverslips in standard media (*Olenych et al., 2007*). Cells are allowed to recover for 48 hr and then fixed with paraformaldehyde (4%) one hour following treatment with FCCP (1 µM) or DMSO. High magnification images of cells are captured by confocal microscopy (DM6, Leica, Wetzlar, Germany) and mitochondrial morphological characteristics were assessed with the Mitochondrial Network Analysis (MiNA) toolset in J-image as previously described (*Valente et al., 2017*; *Martín-Maestro et al., 2017*). Briefly, the plugin converts confocal images to binary pixel features and analyzes the spatial relationship between pixels. The parameters analyzed are: (i) individual mitochondrial structures; (ii) networked mitochondrial; and (iii) the average of length of rods/branches. Twenty randomly chosen fields containing 30–50 cells were used to quantify the morphological pattern and network branch lengths of mitochondria. We classify the mitochondrial morphology as fragmented when the appearance is completely dotted with branch lengths < 1.8 µm.

## Electrophoresis and immunoblot analysis

For SDS-PAGE, Laemmli loading buffer (Bio-Rad Lab, Hercules, CA, USA) plus 5% β-mercaptoethanol was added to protein extracts from cell pellets reconstituted in cell lysis buffer (Cell Signaling Technology, Danvers, MA) before heating at 95°C for 5 min. Around 30 μg of whole cell protein lysate samples were separated on 4–15% mini-PROTEIN GTX precast gels, and transferred to nitrocellulose membranes (Bio-rad). For native electrophoresis, 20 μg of mitochondrial protein extracts were resuspended in buffer containing 50 mM NaCl, 2 mM 6-aminohexanoic acid, 50 mM imidazole, 1 mM EDTA (pH 7), solubilized with digitonin (2 g/g protein) for 20 min on ice, and centrifuged for 20 min at 30,000 g to remove cell debris. Supernatants were removed and 10% glycerol and 0.01% Ponceau S were added as previously described (*Kovalčíková et al., 2019*; *Wittig and Schägger, 2009*). Samples along with a high molecular weight native marker (GE Healthcare Life Sciences, Marlborough, MA) were separated on 4–15% precast gels in 4°C with current limited to 15 mA and transferred to polyvinylidene difluoride membranes (*Wittig and Schägger, 2009*). Immunoblotting was performed after blocking in TBS (Tris-buffered saline) containing 5% non-fat milk and 0.1% Tween-20. Membranes were incubated overnight with primary antibodies diluted in blocking solution at 4°C, followed by incubation with secondary antibodies at room temperature for 1 hr. Immunoreactivity was visualized by the ECL chemiluminescence system (Bio-rad) on standard film. The antibodies were ATP5A (ab-14748, 1:1000, Abcam), ATP Synthase C-subunit (ab-181243, 1:1000, Abcam), and citrate synthase (#14309, 1:1000, Cell Signaling Technology).

## Immunofluorescence and confocal microscopy

For immunocytochemistry of mammalian cells, AGS and mouse NSC/NPC cells were seeded on laminin-coated coverslips (Neuvitro, Vancouver, WA, USA) within 24-well plates. The cells were fixed with 4% paraformaldehyde in PBS, washed with PBS, and permeabilized with 0.02% Triton X-100 in PBS for 10 min. Blocking was done with 5% BSA in PBS for 1 hr, followed by incubation with antibodies against Nestin (MAB2736, 1:50, R and D Systems, Cambridge, MA, USA) or Ki-67 (NB600-1252, 1:500, Novus Biologicals, Littleton, CO, USA) in blocking buffer overnight at 4°C. The Nestin antibody was detected using goat anti-mouse AlexaFluor 488 or 647 (1:1000; Jackson ImmunoResearch Laboratories Inc, West Grove, PA, USA) and the Ki-67 antibody was detected using AlexaFluor 488 goat anti-rabbit (1:1000; Jackson Immunoresearch) or Cy3-conjugated donkey anti-rabbit (1:500; EMD Millipore, Burlington, MA, USA) in blocking buffer. Cells were washed with PBS after primary and secondary antibody staining. Stained cells were overlaid with Fluoroshield mounting medium with DAPI (Abcam) to label nucleus. Fluorescence microscopy was performed with a Leica confocal microscope using the following fluorescence filters: DAPI (405 nm excitation); Cy3 (551 nm excitation); AlexaFluor 647 (651 nm excitation); and GFP/AlexaFluor 488 (488 nm excitation). For comparison across conditions, identical light-exposure levels were used.

## Quantitative RT-PCR

RNA was extracted from approximately 200,000 mouse or AGS NPCs per condition according to manufacturer instructions (Quick-RNA MiniPrep kit; Irvine, CA, USA). Total RNA was reverse transcribed into cDNA (Bimake, Houston, TX, USA), and real-time PCR was performed (LightCycler96, Roche, Basel, CHE) with SYBR Green (Thermo Fisher Scientific) as a dsDNA-specific binding dye. Quantitative RT-PCR conditions were 95°C for denaturation, followed by 45 cycles of: 10 s at 95°C, 10 s at 60°C, and 20 s at 72°C. Species-specific primers for each transcript were used (for list see *Table 1*). Melting curve analysis was performed after the final cycle to examine the specificity of primers in each reaction. Relative abundance of each *Atp5g* isoform as a fraction of total *Atp5g* was calculated by ΔΔCT method and normalized to *Rpl27*.

## Statistical analysis

Data were analyzed using GraphPad Prism Software (Graphpad, San Diego, CA) and presented as means ± S.E. unless otherwise specified, with *P*-values calculated by two-tailed unpaired Student's *t*-tests or two-way ANOVA (comparisons across more than two groups) adjusted with the Bonferroni's correction. No randomization or blinding was used and no power calculations were done to detect a pre-specified effect size.

**Table 1.** Species-specific primers used in quantitative RT-PCR.

| Mouse | |
|---|---|
| Rpl27 Forward | ATA AGA ATG CGG CCG CAA GC |
| Rpl27 Reverse | ATC GAT TCG CTC CTC AAA CTT |
| Atp5g1 Forward | TGC AGA CCA CCA AGG CAC TG |
| Atp5g1 Reverse | GGC CTC TGG TCT GCT CAG GA |
| Atp5g2 Forward | CGT CTC TAC CCG CTC CCT GA |
| Atp5g2 Reverse | CTG CAG ACA GCG GAC GAC TC |
| Atp5g3 Forward | GGG CCC AGA ATG GTG TGT GT |
| Atp5g3 Reverse | TGC AGC ACC TGC ACC AAT GA |
| **AGS** | |
| Rpl27 Forward | CTG CCA TGG GCA AGA AGA AA |
| Rpl27 Reverse | AGC AGG GTC TCT GAA GAC AT |
| Atp5g1 Forward | TCC GGC TCT GAT CCG CTG TA |
| Atp5g1 Reverse | GGG AGC TGC TGC TGT AGG AA |
| Atp5g2 Forward | TGC CTG CTC CAG GTT CCT CT |
| Atp5g2 Reverse | GGG ACT GCC AAG CTG CTG AA |
| Atp5g3 Forward | TGA GGC CCA GAA TGG TGA ACG |
| Atp5g3 Reverse | CAG CAC CAG AAC CAG CCA CT |

## Acknowledgements

NSS and MB receive support from the American Heart Association, 18CDA34030443 and 19POST34381071, respectively. DKM receives support from National Institutes of Health grant R01GM117461, Pew Scholar Award, Curci Faculty Scholar Award from the Innovative Genomics Institute, and Packard Fellowship in Science and Engineering. We thank Dr. Judith Kelleher of Neuronascent for AGS NPCs and acknowledge use of sponsored core facilities including the UCSF Laboratory for Cell Analysis (P30CA082103) and the Histology and Light Microscopy Core at the Gladstone Institutes.

## Additional information

### Funding

| Funder | Grant reference number | Author |
|---|---|---|
| National Institute of General Medical Sciences | R01GM117461 | Dengke K Ma |
| Pew Charitable Trusts | Pew Scholar Award | Dengke K Ma |
| David and Lucile Packard Foundation | Fellowship | Dengke K Ma |
| Innovative Genomics Institute | Curci Scholar Award | Dengke K Ma |
| American Heart Association | 18CDA34030443 | Neel S Singhal Meirong Bai |
| American Heart Association | 19POST34381071 | Neel S Singhal Meirong Bai |

The funders had no role in study design, data collection and interpretation, or the decision to submit the work for publication.

## Author contributions
Neel S Singhal, Conceptualization, Data curation, Formal analysis, Investigation, Methodology, Writing - original draft; Meirong Bai, Evan M Lee, Data curation, Formal analysis, Investigation, Methodology; Shuo Luo, Kayleigh R Cook, Investigation, Methodology; Dengke K Ma, Conceptualization, Resources, Data curation, Supervision, Funding acquisition, Investigation, Visualization, Project administration, Writing - review and editing

## Author ORCIDs
Neel S Singhal https://orcid.org/0000-0003-1605-4444
Meirong Bai https://orcid.org/0000-0002-5919-7464
Dengke K Ma https://orcid.org/0000-0002-5619-7485

## Decision letter and Author response
Decision letter https://doi.org/10.7554/eLife.55578.sa1
Author response https://doi.org/10.7554/eLife.55578.sa2

# Additional files

## Supplementary files
• Supplementary file 1. Cytoprotective AGS genes identified from AGS cDNA screens in mouse NPCs exposed to hypoxia, hypothermia, or rotenone.

• Supplementary file 2. Ground squirrel-unique amino acid substitutions identified from the cytoprotective screened genes and associated Jensen-Shannon Divergence and BLOSUM-62 scores.

• Transparent reporting form

## Data availability
Data has been made available on Dryad (https://doi.org/10.7272/Q6MW2FCP) and code as been made available on GitHub (https://github.com/evanmlee/MaLab_spec_subs; copy archived at https://github.com/elifesciences-publications/MaLab_spec_subs).

The following dataset was generated:

| Author(s) | Year | Dataset title | Dataset URL | Database and Identifier |
|---|---|---|---|---|
| Singhal N, Ma D | 2020 | Data for: Cytoprotection by a naturally occurring variant of ATP5G1 in Arctic ground squirrel neural progenitor cells | https://doi.org/10.7272/Q6MW2FCP | Dryad Digital Repository, 10.7272/Q6MW2FCP |

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
