## [Decision Letter]

**Acceptance summary:**

This work demonstrates an interesting phenomenon that underlies intrinsic tolerance of squirrels to metabolic stressors during hibernation, such as severe hypoxia and low temperature. Through cell survival-based cDNA expression screens and comparative genomics, a variant of ATP5G1 gene, coding for the mitochondrial F-ATP synthase subunit c, is found to have a broad protective effect on the cellular level. Specifically, a single amino acid substitution in mitochondrial targeting sequence of ATP5G1, contributes to the cellular stress resilience by modulating mitochondrial metabolism.

**Decision letter after peer review:**

Thank you for submitting your article "Cytoprotection by a naturally occurring variant of ATP5G1 in Arctic ground squirrels" for consideration by *eLife*. Your article has been reviewed by Diethard Tautz as the Senior Editor, a Reviewing Editor, and three reviewers. The following individuals involved in review of your submission have agreed to reveal their identity: Kelly Drew (Reviewer #1); Jean-Paul di Rago (Reviewer #2); Michela Carraro (Reviewer #3).

The reviewers have discussed the reviews with one another and the Reviewing Editor has drafted this decision to help you prepare a revised submission.

As the editors have judged that your manuscript is of interest, but as described below that additional experiments are required before it is published, we would like to draw your attention to changes in our revision policy that we have made in response to COVID-19 (https://elifesciences.org/articles/57162). First, because many researchers have temporarily lost access to the labs, we will give authors as much time as they need to submit revised manuscripts. We are also offering, if you choose, to post the manuscript to bioRxiv (if it is not already there) along with this decision letter and a formal designation that the manuscript is 'in revision at *eLife*'. Please let us know if you would like to pursue this option. (If your work is more suitable for medRxiv, you will need to post the preprint yourself, as the mechanisms for us to do so are still in development.)

The manuscript reports on an extreme hypoxia resilience observed in the Arctic ground squirrels (AGS). These animals can tolerate hypoxic insults to survive during hibernation, and this phenomenon is maintained by the cultured cells. The work identifies variant ATP5G1 as a protein responsible for cytoprotection against hypoxia. The findings are overall very interesting and important. The work on non-model animals is a strength of this paper. At the same time, the referees realize limitations in experimental technology.

The points included in the specific comments of the reviewers should be addressed to strengthen the data in general and provide more precise explanations. The reviewers were overall not fully satisfied about the level of mechanistic understanding. Thus, the revisions should aim to provide some clarifications and/or discussions on the molecular and mechanistic basis of the ATP5G1 involvement in this resilience phenomenon.

Reviewer #1:

Some weaknesses are noted. A section should be added to the Discussion to identify weakness to be considered if additional information cannot be added to remedy the concern.

1) No randomization or blinding was used and no power calculations were done to detect a pre-specified effect size. Due to the nature of the initial unbiased approach I do not feel this is a fatal flow. However, the results would be more compelling if studies of the effect of amino acid substitutions had been blinded and guided by a prior power analysis. This weakness is clearly acknowledged in the text and should be noted in the discussion.

2) The study is based on NPCs which may not reflect properties of fully differentiated or adult neurons.

3) Further supplemental data or additional references are needed to allow others to replicate the approach used to identify unique amino acid substitutions. In particular what rationale was used to define the red quadrant (high JSD values and low BLOSUM62 scores) in Figure 2D?

4) Error bars should be defined in all figures or figure captions.

Neural progenitor cells should be noted in the title and the abstract.

Reviewer #2:

If I am correct the authors consider that the much higher oxygen consumption rate after FCCP addition in AGS versus mouse NPCs contributes to the differences in survival between these two types of cells when exposed to metabolic stress. Could the authors evoke a possible explanation for this? It would be interesting to known what is responsible for the large difference in uncoupled respiration between AGS and mouse NPCs. Is it because of differences in the levels of respiratory chain complexes? If this difference in respiratory capacity has a cytoprotective effect, how can we explain that after blocking the respiratory chain by rotenone during two days, AGS NPCs still have an enhanced rate of survival than mouse NPCs. This seems to this reviewer contradictory. This point should be clarified.

It is shown that the Pro to Leu variant in the MTS of ATP5G1 in AGS does not compromise import, accumulation and assembly of its protein product and activity of ATP synthase. Thus, this variant does apparently have no consequence on ATP synthase. It is puzzling how this variant could modulate the energy-transduction activity of mitochondria. The suggestion that the MTS with the Pro to Leu variant would remain associate after cleavage with ATP synthase, like the MTS of the Rieske protein in mammals, seems very unlikely in the light of recently published high resolution structures of ATP synthases from various origins. It would be interesting to test the cytoprotective properties of the ATP5G1's MTS, alone or fused to another protein unrelated to ATP synthase.

Reviewer #3:

The authors state that growth rate of AGS and mouse NPCs is not changed although this aspect is not appreciable from a single immunofluorescence image (Figure 1A). The authors also state to have measured cell death upon FCCP stimulation, even though data are not reported (Figure 1B). The Seahorse experiment in Figure 1C poses a serious concern. In general, I think it is hard to make comparisons among cell cultures deriving from different species (here AGS versus murine NPCs) and this would require a number of control experiments that have not been provided in this study. To my knowledge, the fact that mouse NPCs display no respiratory spare capacity is unusual. FCCP is used to push the respiratory chain to its maximal activity, however the amount of this compound should be carefully titrated in order to avoid secondary toxic effects that compromise mitochondrial function. The increased spare capacity visible in AGS NPCs is obtained with 2 μM FCCP, which appears to be ineffective and even detrimental for mouse NPCs. Did the authors test different concentrations of FCCP? Another concern regards the amount of mitochondria that could differ among species. From confocal analysis it emerges indeed that AGS NPCs contain probably fewer mitochondria than their murine counterparts. Another lacking information is the evaluation of the OXPHOS protein level which would help the interpretation of data. Then, the authors evaluated changes in mitochondrial morphology by using FCCP, although they never tested the mitochondrial depolarization upon FCCP stimulation and this is a crucial piece of information that should have been assessed. Anyway, in general I believe that the electron microscopy is a more reliable technique to evaluate mitochondrial morphology in the various species and conditions.

In Figure 2—figure supplement 1E, the authors show the level of ATP5G protein without reporting any other subunits of the F-ATP synthase. Since the generation of a single F-ATP synthase monomer is a multiple-step process which requires a fine-tuned incorporation of different subcomplexes, the alteration (over-expression or down-regulation) of any F-ATP synthase subunit could deeply alter the whole assembly. I believe that this aspect should have been carefully evaluated in the presented models. Moreover, for an accurate measurement of the mitochondrial ATPase hydrolytic activity, the addition of its selective inhibitor oligomycin is mandatory, and its effect should have been used as internal control. The overexpression of AGS and human variants and their mitochondrial localization should have been checked, since alterations in the mitochondrial targeting sequence might affect a correct targeting to the organelle. Figure 2—figure supplement 1F is not cited.

Figure 3—figure supplement 1(panels E and F are not present but are cited in the text) does not correspond to what the authors state (subsection “A cDNA library expression screen identifies AGS ATP5G1 as a cytoprotective factor”). The authors mention that AGS ATP5G1 substitution did not alter mitochondrial localization although this has never been tested. There is instead a single image showing the signal of wild-type ATP5G1 and Cox8 without calculating any colocalization index. The Seahorse experiment in Figure 3A is also puzzling. From the traces, it appears that the overexpression of human ATP5G1 can improve the spare capacity of murine NPCs per se and that the substitution of human ATP5G1 L32P did not further increase the FCCP-stimulated respiration but rather caused a slight decrease. This effect on mitochondrial respiration does not correlate with the resistance observed against cell death shown in Figure 2E-G. This means that the cytoprotection does not rely on improvement of mitochondrial function, thus the molecular explanation remains to be addressed. It is also not clear whether the decreased FCCP-stimulated respiration in AGS ATP5G1 L32P is statistically significant, since no details of number of experiments and statistics have been provided. The confocal experiments then lack the control images of untreated cells to appreciate changes upon FCCP stimulation (and again the mitochondrial membrane depolarization should have been checked).

Concerning the last part on ABE KI cells, I found differences of mean branch length marginal and of a questionable biological meaning. Again, electron microscopy images of mitochondrial morphology would have improved the analysis and the comparison between the two genotypes. Moreover, the correct targeting of the variant ATP5G1 needed to be addressed, since the only western blot showing ATP5G level has been carried out with cell lysates and appears over saturated.

[Editors' note: further revisions were suggested prior to acceptance, as described below.]

Thank you for re-submitting your article "Cytoprotection by a naturally occurring variant of ATP5G1 in Arctic ground squirrel neural progenitor cells" for consideration by *eLife*. Your article has been reviewed by Diethard Tautz as the Senior Editor, a Reviewing Editor, and three reviewers. The following individuals involved in review of your submission have agreed to reveal their identity: Kelly Drew (Reviewer #1); Jean-Paul di Rago (Reviewer #2); Michela Carraro (Reviewer #3).

The reviewers have discussed the reviews with one another and the Reviewing Editor has drafted this decision to help you prepare a revised submission.

In principle, the reviewers are satisfied with the additional experimental work, clarification and text revisions. There is one remaining issue specified below to be addressed prior to the acceptance.

Essential revisions:

Among the newly provided data, there is one aspect that should be carefully revised. The assembly of the F-ATP synthase has been evaluated by a clear-native PAGE and the quantification analysis shows that the D/M ratio is decreased in the ABE KI. The authors state that this is due to a decreased content of monomers, although this is not at all clear in Figure 4J (I found instead a higher level of monomers in ABE KI by looking at ATP5A signal) and it is anyway pretty in contradiction with a decreased D/M ratio. The difference among the two genotypes, which show a comparable amount of monomers, might be instead in the fraction of dimers which appears slightly decreased in the ABE KI (looking at ATP5G1 signal). I believe that a more informative analysis could be perhaps normalizing the fraction of monomers and dimers to the total amount of the F-ATP synthase, e.g. D/(M+D). The authors should revise Results section and Discussion section of this part.

---

## [Author Response]

Reviewer #1:Some weaknesses are noted. A section should be added to the Discussion to identify weakness to be considered if additional information cannot be added to remedy the concern.1) No randomization or blinding was used and no power calculations were done to detect a pre-specified effect size. Due to the nature of the initial unbiased approach I do not feel this is a fatal flow. However, the results would be more compelling if studies of the effect of amino acid substitutions had been blinded and guided by a prior power analysis. This weakness is clearly acknowledged in the text and should be noted in the discussion.

Thank you for your helpful suggestions. We have added additional points to clarify sources of potential false positive and negative candidates from the genetic screen approach in subsection “A cDNA library expression screen identifies AGS ATP5G1 as a cytoprotective factor”. We acknowledge that no blinding or a priori power analysis was performed, and we plan to perform power analyses/blinding procedures as key steps in performing future preclinical in vivo work.

2) The study is based on NPCs which may not reflect properties of fully differentiated or adult neurons.

This is certainly accurate. We have noted this in the Title and Abstract.

3) Further supplemental data or additional references are needed to allow others to replicate the approach used to identify unique amino acid substitutions. In particular what rationale was used to define the red quadrant (high JSD values and low BLOSUM62 scores) in Figure 2D?

We have uploaded the code for the algorithm we produced to Github and indicated this in the data transparency section. This script will allow users to replicate our findings, and in addition, modify species inputs for the identification of unique amino acid substitutions in other organisms of interest. The red quadrant reflects the top 1% of combined JSD and BLOSUM scores, which we clarified in the legend.

4) Error bars should be defined in all figures or figure captions.Neural progenitor cells should be noted in the title and the abstract.

Thank you for noticing this. We have now defined the standard error mean bars in all the captions.

Reviewer #2:If I am correct the authors consider that the much higher oxygen consumption rate after FCCP addition in AGS versus mouse NPCs contributes to the differences in survival between these two types of cells when exposed to metabolic stress. Could the authors evoke a possible explanation for this? It would be interesting to known what is responsible for the large difference in uncoupled respiration between AGS and mouse NPCs. Is it because of differences in the levels of respiratory chain complexes?

This is an important question in the field as the specific OXPHOS proteins and associated factors contributing to cellular reserve respiratory capacity are not completely unknown. The specific mechanism for AGS cells will require additional research, but we have added data that begins to address this and is also related to reviewer #3 concerns (Figure 1—figure supplement 1H). The literature has suggested that complex II OXPHOS proteins are crucial to this response (Pfleger et al., 2015), however, as shown in Figure 1—figure supplement 1H, there is no appreciable difference in complex II activity between AGS/mouse NPCs. Citrate synthase (a marker of mitochondrial content) or complex I activity are also similar between AGS and mouse NPC mitochondria. We did find a decrease in the activity of Complex IV. The contributions of CIV and its supramolecular organization as a mediator of differences between species warrants follow-up and is part of another collaborative manuscript currently in revision.

If this difference in respiratory capacity has a cytoprotective effect, how can we explain that after blocking the respiratory chain by rotenone during two days, AGS NPCs still have an enhanced rate of survival than mouse NPCs. This seems to this reviewer contradictory. This point should be clarified.

This is also an important point. Rotenone results in cell death via oxidative stress and ATP deficiency (Yadava and Nicholls, 2007). AGS likely possess numerous adaptations in these pathways conferring a survival advantage. Our over-expression screen and follow-up ATP5G1 studies suggest that ATP5G1 is one component contributing to this metabolic resilience, but it is likely only one adaptation among many. Other AGS OXPHOS adaptations may also confer survival advantages which help cells effectively bypass complex I and cope with rotenone-induced energy failure. Previous work on hibernating species has also implicated robust anti-oxidant adaptations (Bhowmick and Drew, 2017) in conferring cytoprotection, which is likely also contributing to the AGS phenotype. We have elaborated these points (Discussion section).

It is shown that the Pro to Leu variant in the MTS of ATP5G1 in AGS does not compromise import, accumulation and assembly of its protein product and activity of ATP synthase. Thus, this variant does apparently have no consequence on ATP synthase. It is puzzling how this variant could modulate the energy-transduction activity of mitochondria. The suggestion that the MTS with the Pro to Leu variant would remain associate after cleavage with ATP synthase, like the MTS of the Rieske protein in mammals, seems very unlikely in the light of recently published high resolution structures of ATP synthases from various origins. It would be interesting to test the cytoprotective properties of the ATP5G1's MTS, alone or fused to another protein unrelated to ATP synthase.

We agree that it is unlikely the ATP5G1 MTS remains a part of ATP synthase. We further probed the activity of ATP synthase and along with suggestions from reviewer 3 have added new data demonstrating that ABE KI NPCs have reduced abundance of the monomeric form of ATP synthase under native gel conditions (Figure 4J-K). This indicates that the P32L variant in the MTS actually does compromise the normal assembly of ATP synthase complex. As noted by Vives-Bauza et al., 20170 the ATP5G1 MTS is protective independently of fusion to the mature ATP5G1 peptide, and the possibility that this contributes to the observed phenotypes remains (elaborated on in the Discussion section). Finally, we have preliminarily found binding partners of the N-terminal sequence using mass spectrometry and co-immunoprecipitation. These direct protein interactions likely regulate ATP5G1 stability and processing, but will require further functional validation that we plan to include in follow-up work.

Reviewer #3:The authors state that growth rate of AGS and mouse NPCs is not changed although this aspect is not appreciable from a single immunofluorescence image (Figure 1A).

Thank you for suggesting this. We now show the proliferation rate data in Figure 1B.

The authors also state to have measured cell death upon FCCP stimulation, even though data are not reported (Figure 1B).

Thank you for pointing this out. We’ve removed this incorrect reference.

The Seahorse experiment in Figure 1C poses a serious concern. In general, I think it is hard to make comparisons among cell cultures deriving from different species (here AGS versus murine NPCs) and this would require a number of control experiments that have not been provided in this study. To my knowledge, the fact that mouse NPCs display no respiratory spare capacity is unusual. FCCP is used to push the respiratory chain to its maximal activity, however the amount of this compound should be carefully titrated in order to avoid secondary toxic effects that compromise mitochondrial function. The increased spare capacity visible in AGS NPCs is obtained with 2 μM FCCP, which appears to be ineffective and even detrimental for mouse NPCs. Did the authors test different concentrations of FCCP?

We agree that comparisons between cell type and species types require careful control and interpretation. We previously performed a FCCP dose titration which is now included in Figure 1—figure supplement 1F-G. Also note, we have added the following sentence and references to the Discussion section to further clarify the limited mouse NPC spare respiratory capacity, “…human and mouse NPCs and neural cells have diminished spare respiratory capacity as they may respire maximally at baseline.” We refer to prior reports corroborating this in human and mouse cells: Lorenz et al., 2017, Figure 2C, E; Khacho et al., 2016, Figure 4.

Another concern regards the amount of mitochondria that could differ among species. From confocal analysis it emerges indeed that AGS NPCs contain probably fewer mitochondria than their murine counterparts. Another lacking information is the evaluation of the OXPHOS protein level which would help the interpretation of data.

We also agree that interspecies comparisons are difficult to standardize. The amount of mitochondria and specific make-up of subunits are important to OXPHOS enzymatic activities, and as such we provide additional information regarding mitochondrial and OXPHOS enzymatic function with citrate synthase and individual OXPHOS complex enzymatic activity (Figure 1—figure supplement 1H).

Then, the authors evaluated changes in mitochondrial morphology by using FCCP, although they never tested the mitochondrial depolarization upon FCCP stimulation and this is a crucial piece of information that should have been assessed.

Thank you for suggesting including this important control. We now show previously obtained control data on the effect of FCCP on mitochondrial potential as measured by TMRE (Figure 1E).

Anyway, in general I believe that the electron microscopy is a more reliable technique to evaluate mitochondrial morphology in the various species and conditions.

This will indeed be an extremely valuable contribution. We have embarked on a collaboration to obtain EM data, however, we hope this will be part of a follow-up manuscript related to additional mechanistic details of ATP5G1.

In Figure 2—figure supplement 1E, the authors show the level of ATP5G protein without reporting any other subunits of the F-ATP synthase. Since the generation of a single F-ATP synthase monomer is a multiple-step process which requires a fine-tuned incorporation of different subcomplexes, the alteration (over-expression or down-regulation) of any F-ATP synthase subunit could deeply alter the whole assembly. I believe that this aspect should have been carefully evaluated in the presented models. Moreover, for an accurate measurement of the mitochondrial ATPase hydrolytic activity, the addition of its selective inhibitor oligomycin is mandatory, and its effect should have been used as internal control. The overexpression of AGS and human variants and their mitochondrial localization should have been checked, since alterations in the mitochondrial targeting sequence might affect a correct targeting to the organelle. Figure 2—figure supplement 1F is not cited.

Thanks for this very helpful suggestion. Interestingly, although we found AGS ATP synthase activity to have a similar degree of sensitivity to oligomycin compared to mouse and ABE KI cells, we followed up this finding with clear native electrophoresis and immunoblots demonstrating that ABE WT have decreased abundance of ATP Synthase monomers (Figure 4J-K). Given the extreme complexity of ATP synthase biogenesis and assembly, additional work in the future is still needed to understand how the AGS amino acid substitution alters ATP synthase assembly. Recent work by others and our own preliminary data, which needs further validation, suggest that the N-terminal may be regulated by ubiquitination and also bind to the membrane assembly factor, TMEM70.

Figure 3—figure supplement 1(panels E and F are not even present but are cited in the text) does not correspond to what the authors state (subsection “A cDNA library expression screen identifies AGS ATP5G1 as a cytoprotective factor”). The authors mention that AGS ATP5G1 substitution did not alter mitochondrial localization although this has never been tested. There is instead a single image showing the signal of wild-type ATP5G1 and Cox8 without calculating any colocalization index.

In Figure 3—figure supplement 1, we now provide confocal images of mouse and AGS NPCs expressing all versions of ATP5G constructs. The images demonstrate proper targeting of the AGS and human ATP5G1 constructs to the mouse and AGS NPC mitochondria.

The Seahorse experiment in Figure 3A is also puzzling. From the traces, it appears that the overexpression of human ATP5G1 can improve the spare capacity of murine NPCs per se and that the substitution of human ATP5G1 L32P did not further increase the FCCP-stimulated respiration but rather caused a slight decrease. This effect on mitochondrial respiration does not correlate with the resistance observed against cell death shown in Figure 2E-G. This means that the cytoprotection does not rely on improvement of mitochondrial function, thus the molecular explanation remains to be addressed. It is also not clear whether the decreased FCCP-stimulated respiration in AGS ATP5G1 L32P is statistically significant, since no details of number of experiments and statistics have been provided. The confocal experiments then lack the control images of untreated cells to appreciate changes upon FCCP stimulation (and again the mitochondrial membrane depolarization should have been checked).

We clarified the statistical details in the text and figure caption. Statistically there was no difference between the spare respiratory capacity in mouse NPCs expressing human ATP5G1, agsATP5G1-L32P or the human ATP5G1-P32L. Thus, it does appear that the survival to metabolic stressors does not correlate perfectly with the spare respiratory capacity. Cell survival from metabolic stresses likely engages additional cellular mechanisms as it is different from the acute changes measured in the Seahorse assay. We now discuss that additional mechanisms beyond spare respiratory capacity could be involved on subsection “A cDNA library expression screen identifies AGS ATP5G1 as a cytoprotective factor”. In addition, with the added experiments in response to reviewer 2 comments, we provide a discussion of additional mechanisms contributing to the survival phenotype in the Discussion section. We have added representative control confocal images to the top panels of Figure 3D-G and left panels of Figure 4E,F.

Concerning the last part on ABE KI cells, I found differences of mean branch length marginal and of a questionable biological meaning. Again, electron microscopy images of mitochondrial morphology would have improved the analysis and the comparison between the two genotypes.

The mean mitochondrial network branch length calculated using automated software from confocal images has been used to assess mitochondrial dynamics on populations of cells in response to treatments and helps corroborate other aspects of mitochondrial morphology (fragmented vs elongated mitochondria). Mutations in master regulators of fission and fusion cause larger changes in mitochondrial length in cells at baseline (up to 0.5-1 mm in some reports), however, we feel that the population-based morphological changes in response to FCCP as well as the mean branch length changes present in our experiments (<0.5 mm) are still an important reflection of the mitochondrial phenotype as related to cell genotype and treatment. Nonetheless, we have moved some of the mean branch length data to the supplemental figures in favor or new figures above. As above we whole-heartedly agree with the utility of EM data and plan to obtain this for follow-up reports.

Moreover, the correct targeting of the variant ATP5G1 needed to be addressed, since the only western blot showing ATP5G level has been carried out with cell lysates and appears over saturated.

We now provide additional mitochondrial localization data in Figure 3—figure supplement 1 showing correct targeting of various ATP5G1 constructs to the mouse NPC mitochondria in AGS NPCs expressing co-localized ATP5G1 variants and the Cox8 mitochondrial marker.

[Editors' note: further revisions were suggested prior to acceptance, as described below.]

Reviewer commentsAmong the newly provided data, there is one aspect that should be carefully revised. The assembly of the F-ATP synthase has been evaluated by a clear native PAGE and the quantification analysis shows that the D/M ratio is decreased in the ABE KI. The authors state that this is due to a decreased content of monomers, although this is not at all clear in Figure 4J (I found instead a higher level of monomers in ABE KI by looking at ATP5A signal) and it is anyway pretty in contradiction with a decreased D/M ratio. The difference among the two genotypes, which show a comparable amount of monomers, might be instead in the fraction of dimers which appears slightly decreased in the ABE KI (looking at ATP5G1 signal). I believe that a more informative analysis could be perhaps normalizing the fraction of monomers and dimers to the total amount of the F-ATP synthase, e.g. D/(M+D). The authors should revise Results section and Discussion section of this part.

Thank you for drawing our attention to this important oversight. We also agree that our text was incongruent with the data presented and have modified both as suggested. The clear native PAGE data demonstrates a reduction in the dimerization in the ABE KI cell line which we clarify in subsection “Knock-in of AGS ATP5G1L32P alters the resilient phenotype of AGS cells” and the Discussion section. We have also altered the quantification chart (Figure 4K) to reflect Dimer/Total ATP Synthase (Monomer + Dimer), which also supports the conclusion that dimerization is reduced in the ABE KI cell line.

We have also explicitly stated that conclusions relevant to the mechanistic details of AGS ATP5G1 cytoprotection require additional supporting data (subsection “Knock-in of AGS ATP5G1L32P alters the resilient phenotype of AGS cells”; Discussion section).